

# Detection of $O_4$ absorption around 328 nm and 419 nm in measured atmospheric absorption spectra

Johannes Lampel[1,*], Johannes Zielcke[2], Stefan Schmitt[2], Denis Pöhler[2], Udo Frieß[2], Ulrich Platt[2], and Thomas Wagner[1]

[1]Max Planck Institute for Chemistry, Mainz, Germany
[2]Institute of Environmental Physics, University of Heidelberg, Germany
[*]Now at: Institute of Environmental Physics, University of Heidelberg, Germany

*Correspondence to:* J. Lampel (johannes.lampel@iup.uni-heidelberg.de)

**Abstract.**

Retrieving the column of an absorbing trace gas from spectral data requires that all absorbers in the corresponding wavelength range are sufficiently well known. This is especially important for the retrieval of weak absorbers, whose absorptions are often in the $10^{-4}$ range.

Previous publications on the absorptions of the oxygen dimer $O_2$-$O_2$ (or short: $O_4$) list absorption peaks at 328 nm and 419 nm, for which no spectrally resolved literature cross-sections are available. As these absorptions potentially influence the spectral retrieval of various trace gases, such as HCHO, BrO, OClO and IO, their shape and magnitude needs to be quantified.

    We assume that the shape of the absorption peaks at 328 nm and 419 nm can be approximated by their respective neighboring absorption peaks. Using this approach we obtain estimates for the wavelength of the absorption and its magnitude. Using

Longpath Differential Optical Absorption Spectroscopy (LP-DOAS) observations and Multi-Axis (MAX)-DOAS observations, we estimate the peak absorption cross-sections of $O_4$ to be $(1.7 \pm 0.2) \times 10^{-47}$ cm$^5$ molec$^{-2}$ and determine the wavelength of its maximum at $328.51 \pm 0.15$ nm.

    For the absorption at $419.0 \pm 0.4$ nm a peak $O_4$ cross-section value is determined as $(3.7 \pm 2.7) \times 10^{-48}$ cm$^5$ molec$^{-2}$.

## 1   Introduction

The collision induced absorption of the $O_2$-$O_2$ dimer (or short: $O_4$) needs to be considered in various wavelength regions for in-situ and remote sensing absorption spectroscopy of various atmospheric trace gases. Furthermore, the $O_4$ absorption governs the budget of tropospheric singlet oxygen $O_2(^1\Delta)$ (Schurath, 1986), which can potentially impact on the oxidation of atmospheric trace gases, like CO (Platt and Perner, 1980) and $SO_2$ (Sorokin, 2010). In addition, the absorption of $O_4$ itself can be used to deduce information about the actually observed light paths for passive remote sensing applications by applying

inverse modelling methods and radiative transfer models.

    Ellis and Kneser (1933) reported absorption peaks observed for liquid oxygen at 328 nm and 419 nm (for an overview, see Table 2). Most of the absorption peaks in the UV/Vis range of liquid oxygen can be also found for gas phase oxygen, as measured by Janssen (e.g. 1885); Salow and Steiner (e.g. 1936); Greenblatt et al. (e.g. 1990); Hermans et al. (e.g. 1999);





Thalman and Volkamer (e.g. 2013) and others. It was first observed in the atmosphere by Perner and Platt (1980). These absorptions are potentially shifted by less than a nanometre compared to the liquid phase.

Spectrally resolved cross-section data suitable for spectroscopy applications exists for the absorption peaks at 344, 361, 380, 447, 477, 533, 577 nm and continues further into the red spectral range. Due to instrumental limitations (detection limits and/or covered spectral range), spectrally resolved cross-section data for the absorption peaks at 328 nm (transition $^3\Sigma ^3\Sigma \rightarrow ^1 \Delta ^1\Delta; \nu = 3$, see also Table 2) and 419 nm (transition $^3\Sigma ^3\Sigma \rightarrow ^1 \Delta ^1\Sigma; \nu = 2$) are, to our knowledge, not reported in literature.

Salow and Steiner (1936) measured the intensity of the absorption at 328 nm to be 4.2 % of the $O_4$ absorption at 360 nm, i.e. 15% of the $O_4$ absorption at 344 nm. For an $O_4$ dSCD of $4 \times 10^{43}$ molec$^2$ cm$^{-5}$, as e.g. found in MAX-DOAS observations under low elevation angles, this corresponds to an optical depth of $1.7 \times 10^{-2}$ at 360 nm (Thalman and Volkamer, 2013), and thus an optical depth of $7 \times 10^{-4}$ at 328 nm. This could introduce systematic biases in the spectral retrievals of BrO, HCHO, OClO and $SO_2$, when this wavelength range is included in the respective spectral analysis, as e.g. in satellite retrievals of HCHO (De Smedt et al., 2008), various observations of BrO in the boundary layer (see Vogel et al. (2013) and references therein) and ground based measurements of $SO_2$ (e.g. Schreier et al. (2015) and Wang et al. (2016)). Convoluted to a spectral resolution of 0.5 nm, the overlaying absorption structure of HCHO at 328 nm has a peak absorption cross section of $4.5 \times 10^{-20}$ cm$^2$ molec$^{-1}$ (Chance and Orphal, 2011). Typical ground-based HCHO dSCDs observed on the tropical open ocean are about $1 - 3 \times 10^{16}$ molec cm$^{-2}$ (e.g. Peters et al., 2012) (and lower at higher latitudes, (e.g. De Smedt et al., 2008)), which corresponds to an optical depth of $4 - 14 \times 10^{-4}$ at 329 nm. Therefore it is important to consider the possible $O_4$ absorption at around 328 nm if the retrieval wavelength range for HCHO is extended towards shorter wavelengths below 330 nm. The recommended HCHO setting from Pinardi et al. (2013) suggests a fit interval from 336.5–359 nm and thus does not include the $O_4$ absorption at around 328 nm. This recommendation is for an analysis with fixed Fraunhofer reference spectrum. If sequential references are used (as it is typically done today for the spectral retrieval of weak absorbers, as e.g. in ??), then the spectral range can/should be extended towards shorter wavelengths (e.g. 323 nm).

In addition to that, for the spectral retrieval of BrO, Vogel (2012) found a lower limit for the choice of reliable fit intervals using the method from Vogel et al. (2013) at about 330.6 nm, which could have been caused by the neglected $O_4$ absorption at around 328 nm - the FWHM (Full Width Half Maximum) of the known absorption peaks at 344 and 360 nm amounts to 4 nm. This fit interval was then widely used e.g. in Lübcke et al. (2014); General et al. (2015); Gliß et al. (2015); Bobrowski et al. (2015) and Lübcke et al. (2016). Widening the spectral retrieval interval for both species could reduce the fit error of BrO and HCHO dSCDs, as relatively large absorption bands are found for both species below 330 nm.

For the absorption at 419 nm no estimate for its intensity is known. This spectral region is also known for uncertainties of the available water vapour cross-section as reported in Lampel et al. (2015b), which overlays potential $O_4$ absorptions. Using the same ratio of peak absorption cross-sections of $O_4$ at 477 and 446 nm, an absorption band at 419 nm could potentially have a peak OD of $2 \times 10^{-4}$ for an $O_4$ dSCD of $4 \times 10^{43}$ molec$^2$ cm$^{-5}$. This spectral range is of interest, as it is typically included in the spectral analysis of IO (e.g. Großmann et al., 2013; Prados-Roman et al., 2015). The potential to obtain residual spectra with an RMS below $1 \times 10^{-4}$ has been demonstrated (e.g. Lampel et al., 2015a; Zielcke, 2015) and allows to measure IO at sub-ppt concentrations, corresponding to an OD along 10 km lightpath of $< 7 \times 10^{-4}$. Such concentrations have been reported



on the open ocean by Prados-Roman et al. (2015) and were found above their respective limit of detection by Zielcke (2015) in polar regions (0.2-0.4 ppt using LP-DOAS and MAX-DOAS observations).

In this work, spectral data from three different instruments and environmental settings, utilizing two different remote sensing geometries, is presented in order to estimate the magnitude of the $O_4$ absorption peaks at around 328 and 419 nm. The different

techniques used and the different locations of the measurements make it very unlikely that instrumental artefacts or local atmospheric influences play a role.

## 2 Measurement Campaigns

In this section we shortly describe the measurement campaigns, during which DOAS data was collected and which is used in this manuscript.

### 2.1 MAX-DOAS

### 2.2 Antarctica LP-DOAS 2012

In 2012, a measurement campaign in Antarctica was undertaken based at the New Zealand station Scott Base. The station is situated on Ross Island at 78°S and the campaign lasted throughout austral spring, from the end of August until the end of November.

The LP-DOAS instrument used in this campaign has been used in previous studies and therefore has already been extensively described elsewhere, e.g. in Pöhler et al. (2010); Eger et al. (2017).

In brief, light is coupled in and out of a telescope using a Y-shaped fiber optic bundle consisting of 7 individual fibers. The common end is mounted near the focal point of the telescope, the single central fiber is connected to the spectrometer and the six outer fibers are attached to a 800 μm mono fiber which leads to the light source. A diffuser plate can be driven into the light

path in front of the telescope end of the fiber in order to record spectra of the lamp (so-called optical shortcut spectra). For most of the time of the campaign, a 75 W xenon arc lamp (Osram XBO, not ozone-free) was used as the light source. During the last two weeks, a 500 W arc lamp (PLI Hanovia HSAX5002) was used, which has a higher luminous density and thus leads to a lower limit of detection in the deeper UV region around 300 nm. Both lamps were coupled into the fiber using a single fused silica lens. Besides several color filters to reduce stray light, the lamp housing also featured a shutter in front of the fiber

coupling to block the lamp, in order to be able to record background spectra with the telescope, i.e. spectra of the light entering the telescope due to scattering on the surface or air and not from the dedicated light source. As spectrometer, an Acton 300i was used with a resolution of 0.50 nm and spectra were recorded by a back-illuminated CCD camera from Roper Scientific (Spec-10:2KBUV).

Two light paths were set up approximately 1.5 m over sea ice, for which two retro reflector arrays were deployed. The retro

reflector arrays reflecting the light back into the telescope consisted of 12 (short path) and 50 (long path) individual elements made of fused silica with a diameter of 63.5 mm each. One array was located closer to the station, to be able to measure during



periods with fog or snow drift, the other one further away to achieve lower detection limits. They were located at distances of 1.46 km (short) and 4.01 km (long).

The measurement procedure was as follows. Spectra were acquired alternatingly on both light paths. For each light path, spectra were recorded at four different wavelength intervals. The region of interest for this study is the one between 271 nm and 355 nm. For each of those regions, 25 spectra with a saturation of 70% were recorded, with a maximum duration of 1 s per spectrum. Typical exposure times for one spectrum were between 6 ms and 500 ms depending on the wavelength range, visibility conditions and the used arc lamp.

### 2.2.1 Polarstern 2014–2016

The MAX-DOAS instrument used during these cruises PS88-PS98 (ANT XXX, ARK XXIX, ANT XXXI [1]) onboard R.V. Polarstern from October 2014 to April 2016 is described in Lampel et al. (2017) and the same upper limits for the spectral stability of the instrument also apply here for each leg. As during previous cruises, the exposure time per spectrum was set to two minutes. Spectra were recorded at 7 elevation angles of 90° (zenith), 40, 20, 10, 5, 3, 1°, respectively, as long as solar zenith angles (SZA) were below 85°. To reduce RMS, four elevation sequences were co-added before the DOAS analysis.

### 2.2.2 Penlee Point Atmospheric Observatory 2015–2016

MAX-DOAS measurements were performed from 3 April 2015 to 3 March 2016 at the Penlee Point Atmospheric Observatory (PPAO) on the south-west coast of the UK (e.g. Yang et al. (2016)).

Similar to the instrument used in Lampel et al. (2015a) during the MAD-CAT campaign, the instrument is based on an Avantes ultra-low stray-light AvaSpec-ULS2048x64 spectrometer ($f = 75$ mm) using a back-thinned Hamamatsu S11071-1106 detector. The spectrometer is temperature stabilized ($\Delta T < 0.02\,°$C). The UV spectrometer covered a spectral range of 296–459 nm at a FWHM spectral resolution of $\approx 0.55$ nm (at 334 nm) or $\approx 6$ pixel. The spectral stability was sufficiently high with a diurnal shift of less than $\pm 3$ pm. During the night, mercury discharge lamp spectra were recorded automatically in order to measure the instrument's spectral response function. No significant change of the response function was observed during the campaign.

The elevation sequence included elevation angles of -2°,-1°,1°,2°,3°,5°,10°,20°,40°and 90° heading towards an azimuthal south-westerly direction of 245°. After 22 January 2016, the azimuthal viewing direction was changed to a south-easterly direction of 147°. Spectra with a total exposure time of one minute were recorded at an adaptive integration time per scan in order to obtain spectra with a maximum saturation of 50%. From 5 June to 27 August 2015 the total exposure time per spectrum was reduced to 10 s.

The inherent non-linearity of the measured intensity values with respect to the actual incoming intensity of the spectrometers was corrected for by multiplying all intensities with a non-linearity correction polynomial. This polynomial was determined from a set of spectra recorded at different exposure times, which were recorded previous to the campaign using a temperature stabilized "white" LED light source.

---

[1]The respective cruise reports can be found at https://www.pangaea.de/expeditions/cr.php/Polarstern



In order to have a coherent dataset and to reduce the RMS noise of the fits, spectra of subsequent elevation angle sequences during one day were co-added during preprocessing in order to obtain a consistent MAX-DOAS data set with spectra with a total exposure time of 4 minutes.

## 3  Method

Two DOAS methods were applied to quantify the $O_4$ absorption around 328 nm. The MAX-DOAS measurements have often longer effective light paths, which is however not initially known. LP-DOAS measurements have the advantage that the absolute light path length is known, but often yield larger fit residuals.

### 3.1  MAX-DOAS

The MAX-DOAS elevation sequences were evaluated against a current Fraunhofer reference using the sum of the two nearest
zenith sky spectra in order to minimize the effect of stratospheric absorbers. At the same time, this approach minimizes the effect of instrumental instabilities on the data evaluation.

The literature absorption cross-sections listed in Table 1 were convoluted with the measured Mercury (Hg) emission line at 334 nm. The Ring spectrum was calculated using DOASIS (Kraus, 2006), the correction spectrum for vibrational Raman scattering of molecular nitrogen (VRS,  Lampel et al., 2015a) was calculated from the Fraunhofer reference itself, shifting the
spectrum by the corresponding vibrational energy quantum.

HONO and $SO_2$ were not detected in significant amounts, neither was OClO. Water vapour absorption in the UV as reported in Lampel et al. (2017) is small below 358nm ($\sigma < 3 \times 10^{-28}$ cm$^2$ molec$^{-1}$ ) and thus negligible, especially in polar regions.

### 3.2  LP-DOAS

The recorded spectra from the LP-DOAS were co-added over 32 measurement sequences to achieve a higher signal-to-noise
ratio. The optical depth was then calculated by dividing atmospheric and optical shortcut spectra after their respective background spectra had been subtracted. The optical densities were then high-pass filtered before the fit was applied. The applied fit scenario settings are detailed in Table 1. As for the MAX-DOAS evaluation, the literature cross-sections were convoluted with a measured Hg emission line at 334 nm.

### 3.3  Hypothetical $O_4$ absorption cross-sections

The absorption peak shape of the $O_4$ absorption at 328 and 419 nm cannot be deduced directly from field measurements, as a large number of other absorbers (HCHO, BrO, $SO_2$, HONO, OClO, $NO_2$, $O_3$, $H_2O$ and others) potentially overlay the respective $O_4$ absorption peak. Their abundance is unknown in field observations. Additionally, their absorption cross-sections may not be known precisely enough (e.g. water vapour, Lampel et al. (2015b, 2017)) to determine their abundances in other spectral ranges and thus constrain their overall absorption in order to obtain an extended $O_4$ absorption cross-section from
MAX-DOAS or LP-DOAS field measurements. Dedicated laboratory studies will be needed.





| | | MAX-DOAS | | LP-DOAS | |
|---|---|---|---|---|---|
| | | $O_4$ 328 nm | $O_4$ 419 nm | $O_4$ 477 nm | $O_4$ 328 nm | |
| Wavelength interval | | 322.5/311.5 | 410 | 450 | 320 | |
| nm | | 358 | 439 | 490 | 347.5 | |
| $O_3$ | 223K | × | × | × | × | Serdyuchenko et al. (2014) |
| | 223K | (×) | | | | Taylor expansion terms (Puķīte et al., 2010) |
| | 243K | × | | | | |
| HCHO | | × | | | × | Chance and Orphal (2011) |
| BrO | | × | | | × | Fleischmann (2004) |
| $NO_2$ | 293K | × | × | × | × | Vandaele et al. (1998) |
| $O_4$ | | × | × | × | × | Thalman and Volkamer (2013) |
| $SO_2$ | | (×) | | | | Vandaele et al. (2009) |
| HONO | | (×) | | | | Stutz et al. (2000) |
| OClO | | (×) | | | | Kromminga et al. (1999) |
| $H_2O$ | 298K | | × | × | | HITRAN 2012 (Rothman et al., 2013) |
| | | | | | | corrected (Lampel et al., 2015b) |
| IO | | | × | | | Spietz et al. (2005) |
| Glyoxal | | | (×) | | | Volkamer et al. (2005) |
| Ring Spectrum at 273K | | × | × | × | | DOASIS (Kraus, 2006) |
| Ring Spectrum at 243K | | × | | | | |
| Ring Spectrum $\cdot\lambda^4$ | | × | × | × | | Wagner et al. (2009) |
| VRS ($N_2$) | | (×) | × | | | Lampel et al. (2015a) |
| Background & Shortcut Spectrum | | | | | × | |
| Polynomial degree | | 5 | 5 | 3 | 4 | |
| Add. Polynomial degree | | 1 | 1 | 1 | | |

**Table 1.** Retrieval wavelength intervals for DOAS measurements. Values in brackets were used for sensitivity studies only.

The shape of the $O_4$ absorption peaks at 344, 360, 380 and at 446 and 477 nm is similar as shown in Figure 1 when plotting the absorption cross-section (here from Thalman and Volkamer (2013)) over the difference in wavenumbers to the peak absorption. We therefore guessed the shape of the potential absorption bands at 328 nm and 419 nm by shifting the $O_4$





absorption peaks at 344 and 446 nm by $1414\,\mathrm{cm}^{-1}$ and $1476\,\mathrm{cm}^{-1}$, respectively. These shifts were chosen arbitrarily based on previous publications (Salow and Steiner, 1936; Ellis and Kneser, 1933) which list the wavelength of the respective absorption peaks.

This approach is reasonable, as the width of the $O_4$ absorption peaks is defined by the lifetime of the collision complex (Long and Ewing, 1973; Thalman and Volkamer, 2013) and is thus related to the energy of the respective absorption peak. The absorption peak shape at 477 nm was parametrized by Sneep and Ubachs (2003), but we are not aware of parametrizations or quantum-mechanical calculations yielding absorption peak shapes at other wavelengths at room temperature.

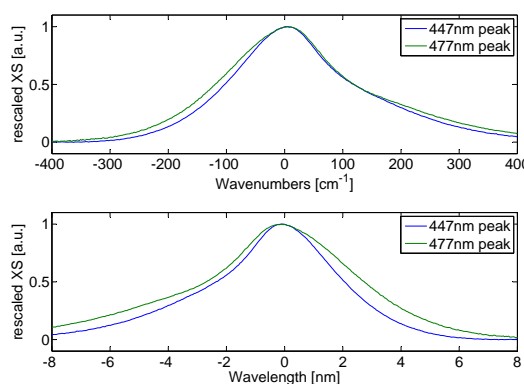

**Figure 1.** Shifted (top: by wavenumber; bottom: by wavelength) and normalized $O_4$ absorption cross-sections at 446 nm and 477 nm according to Thalman and Volkamer (2013).

## 4   Results

### 4.1   Peak positions

As we assume the maximum of the $O_4$ absorption peak at 328.2 nm (Salow and Steiner, 1936) and at 419 nm (Ellis and Kneser, 1933), its exact wavelengths needs to be determined again. This can be done based on experimental MAX-DOAS data, here from PPAO: When included in a DOAS fit, the individual shift of the hypothetical $O_4$ absorption cross-section can be used in order to give a more accurate estimate for the spectral position of the absorption peak. Measurements, which showed a ratio of fitted dSCD and dSCD fit error of more than 8 were considered for this analysis. The resulting peak position was found at

$328.51 \pm 0.15$ nm and thus results in an overall shift of the absorption at 344 nm of $+1366 \pm 13\mathrm{cm}^{-1}$. In the blue wavelength range it was found at $419.32 \pm 0.4$ nm and thus results in an overall shift of the absorption at 446 nm of $+1458 \pm 23\mathrm{cm}^{-1}$. For the absorption around 419 nm a larger scatter of shift values was observed, probably due to uncertainties of the overlaying water vapour absorption.





| Transition | | Wavelength | | | | rel. Intensity | | | |
|---|---|---|---|---|---|---|---|---|---|
| | | [ElKn] | [SaSt] | [ThVo] | This study | [ElKn] | [SaSt] | [ThVo] | This study |
| | | [nm] | [nm] | [nm] | [nm] | [a.u.] | [a.u.] | | [a.u.] |
| $^3\Sigma+^3\Sigma$ | $\rightarrow^1\Delta+^1\Delta(\nu=4)$ | | $315.0\pm0.6$ | | | | | | |
| | $\rightarrow^1\Delta+^1\Delta(\nu=3)$ | $328.9\pm1.5$ | $328.2\pm0.3$ | | $328.51\pm0.15$ | | 0.063 | | $0.18\pm0.02$ |
| | $\rightarrow^1\Delta+^1\Delta(\nu=2)$ | $343.9\pm1.5$ | $343.6\pm0.2$ | $343.8\pm0.1$ | | 10 | 0.42 | 0.95 | 1 |
| | $\rightarrow^1\Delta+^1\Delta(\nu=1)$ | $360.7\pm2$ | $360.7\pm0.2$ | $360.86\pm0.01$ | | 50 | 1.5 | 4.28 | |
| | $\rightarrow^1\Delta+^1\Delta(\nu=0)$ | $380.7\pm2$ | $380.3\pm0.2$ | $380.21\pm0.01$ | | 30 | 0.87 | 2.42 | |
| $^3\Sigma+^3\Sigma$ | $\rightarrow^1\Sigma+^1\Delta(\nu=3)$ | $392.5\pm5$ | | | | | | | |
| | $\rightarrow^1\Sigma+^1\Delta(\nu=2)$ | $419.0\pm1$ | | | $419.02\pm0.42$ | | | | $0.07\pm0.05$ |
| | $\rightarrow^1\Sigma+^1\Delta(\nu=1)$ | $447.0\pm1$ | $446.4\pm0.5$ | $446.39\pm0.01$ | | 4 | 0.15 | 0.53 | 1 |
| | $\rightarrow^1\Sigma+^1\Delta(\nu=0)$ | $476.0\pm2$ | $477.0\pm0.1$ | $476.89\pm0.01$ | | 50 | 2.2 | 6.63 | |
| $^3\Sigma+^3\Sigma$ | $\rightarrow^1\Sigma+^1\Sigma(\nu=4)$ | $462.0\pm1$ | | | | | | | |
| | $\rightarrow^1\Sigma+^1\Sigma(\nu=3)$ | $495.0\pm1.5$ | | | | | | $<0.01$ | |

**Table 2.** Relative intensities of $O_4$ absorption peaks below 500 nm, [SaSt] (Salow and Steiner, 1936) [ElKn] (Ellis and Kneser, 1933) [ThVo] (Thalman and Volkamer, 2013). The column of relative intensities for [ThVo] at a temperature of 293K is given in $10^{-46}$ cm$^5$ molec$^{-2}$. Transitions according to Greenblatt et al. (1990).

## 4.2 Absorption band at 328 nm

### 4.2.1 LP-DOAS

The LP-DOAS data evaluation reveals the suspected absorption structure at 328 nm nicely, given the extensive averaging of the data as described in Section 3.2. An exemplary DOAS fit is shown in Figure 2. Compared to the ozone and $O_4$ absorption, the other species feature only a negligible optical depth in this instance. Besides ozone and the $O_4$ absorption band at 344nm, the 328 nm absorption however is the most prominent absorption feature in this case and spectral region.

In order to compare the relative absorption strength of the 328 nm and 344 nm bands, the entire dataset was filtered for evaluations with a residual RMS better than $1.2\times10^{-4}$ since the structure of interest can only then be retrieved with relative accuracy. Typical optical densities of the 328 nm absorption are between $4-6\times10^{-4}$ along the 8 km light path and column densities reach up to a maximum of around 20 times the instrumental detection limit.

Figure 3 shows the correlation of the LP-DOAS column densities of the suspected 328 nm absorption feature with the well known absorption feature at 344 nm. Data for the short light path is shown in orange, while data for the long path is shown



**Figure 2.** Exemplary LP-DOAS fit result of a measurement from 22nd November 2012 on the 8 km light path. The $O_4$ absorption at 328 nm is clearly visible. The retrieved column densities are: $O_4$ $(2.67 \pm 0.04) \times 10^{43}$ molec$^2$cm$^{-5}$ and $O_4$ 328 nm $(6.09 \pm 0.38) \times 10^{42}$ molec$^2$cm$^{-5}$.



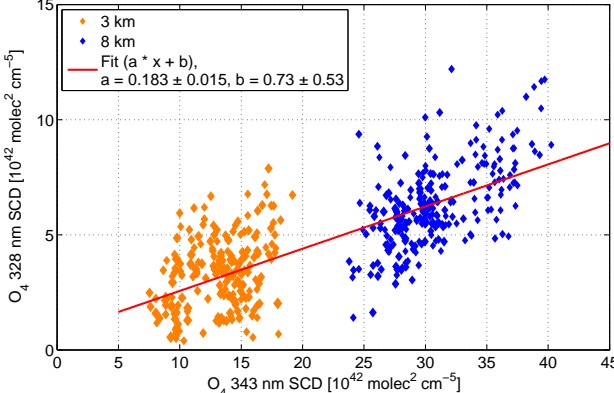

**Figure 3.** LP-DOAS correlation of the suspected 328 nm absorption feature with the known absorption at 344 nm. Data for the short light path is shown in orange, for the long path in blue.

in blue. As can be seen, a correlation between the suspected and the known absorption is visible. As expected, higher column densities for the longer light path and lower ones for the shorter light path. A linear fit performed on the entire dataset indicates a relative absorption strength of $0.183 \pm 0.015$ (see also Table 3) compared to the known 344 nm band.

A variation of the total $O_4$ SCD along both light paths can be observed, which is found simultaneously for both $O_4$ absorption

bands at 328 nm and 344 nm, leading to the distribution of the measurements shown in Figure 3. This points towards a general property of the absorber $O_4$ and can be explained by temperature variations (up to 35K within the measurement period) and pressure variations. We estimate that almost two thirds of the observed variation are due to the changing absorption cross-section with temperature (Thalman and Volkamer, 2013), one third due to the change in number density according to the ideal gas law. Pressure variations between 962 and 1006 hPa during the observations also contribute. Variations depending

on temperature and pressure were also reported by Wang et al. (2016) and Wagner (2017) for ground-based MAX-DOAS observations and simulated for satellite observations by Park et al. (2017) and Dörner et al. (2017). A further and more detailed analysis of these dependencies is not within the scope of this publication.

### 4.2.2 MAX-DOAS

For the PPAO dataset, several sensitivity studies were performed in order to estimate the effect of ozone absorption and the

15 contribution of vibrational Raman scattering (VRS) on the results.

To account for non-linear effect of strong ozone absorption, we also included a wavelength-scaled version of the ozone absorption as well as its square term as suggested in Pukīte et al. (2010) and Pukīte and Wagner (2016). This was mainly necessary when extending the fit interval below 320 nm to include additional absorption bands of HCHO. One part of the absorption of ozone is due to changes in the slant stratospheric ozone column during the recording of the elevation sequence

and thus assumed to be symmetric and not introduce a systematic bias on the evaluation of the $O_4$ absorption band. The tropospheric contribution is of similar magnitude.





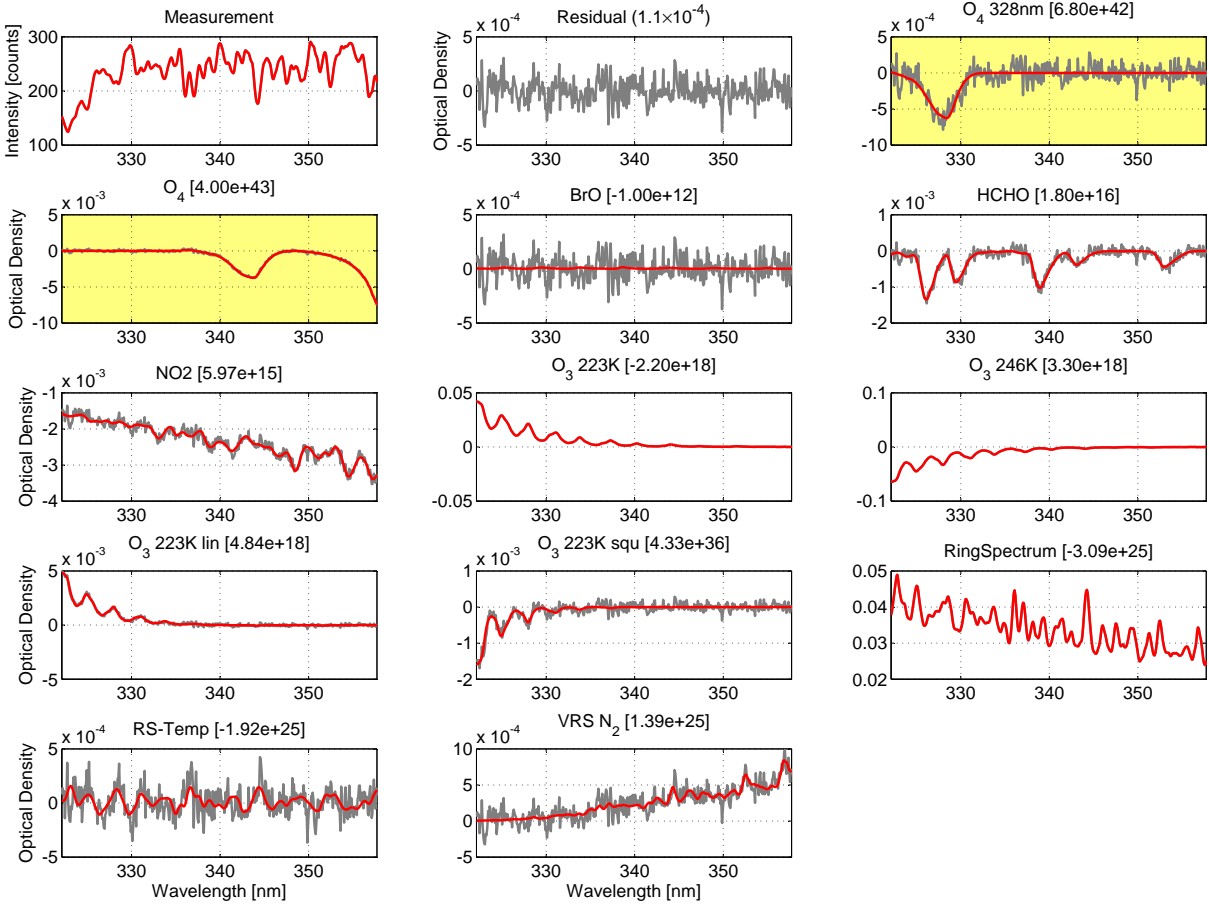

**Figure 4.** MAX-DOAS fit from PPAO, 6/7/2015, 11:18 UTC at $3°$ elevation and a $90°$ elevation reference spectrum. Spectra were co-added to obtain a total exposure time per spectrum of 4 minutes. The $O_4$ absorption structure is found here at a dSCD which is 14 times as large as the DOAS fit error. The plot for the residual shows the residual in grey for the fit considering the $O_4$ absorption at 328nm. Without considering this absorption, the HCHO dSCD in this case is larger($+2.5 \times 10^{14}$ molec cm$^{-2}$ or 1.4%), as is the RMS of the residual ($+2.5 \times 10^{-5}$ or +22%)





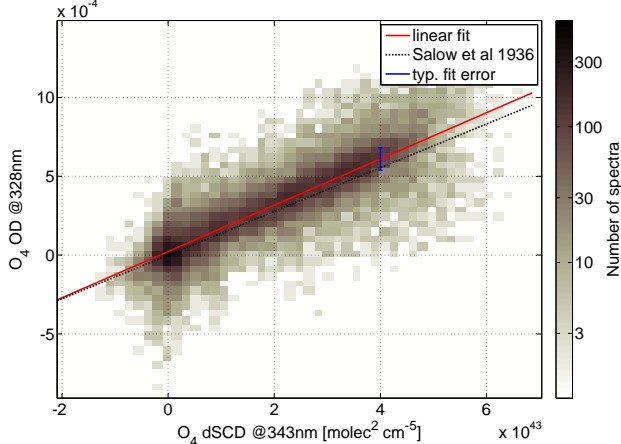

**Figure 5.** Correlation of $O_4$ dSCD at 344 nm and the fitted $O_4$ OD at 328.5 nm assuming the same shape of the absorption cross-section as at 344 nm, using the PPAO dataset.

The additional intensity caused by VRS did not significantly change the result as seen from Table 3.

BrO was analysed in the spectral range from 332–358 nm and not found above the detection limit ($1 \times 10^{13}$ molec cm$^{-2}$) at PPAO. During the Polarstern cruises the amount of HCHO (evaluated in the same spectral range) was typically smaller than at PPAO, however, also significant amounts of BrO were observed during Arctic spring (May, June, up to $1.4 \times 10^{14}$ molec cm$^{-2}$ at $3^\circ$ elevation) and Antarctic spring (December, January, up to $5 \times 10^{13}$ molec cm$^{-2}$ at $3^\circ$ elevation). We therefore exclude the possibility that the absorption at 328 nm is caused by erroneous BrO or HCHO absorption cross-sections.

As the MAX-DOAS measurements took place in regions without strong anthropogenic pollution, tropospheric $NO_2$ absorption does not significantly contribute to the overall magnitude of the observed residual spectra.

The resulting estimates for the absorption cross-section of $O_4$ at 328 nm can be found in Table 3. Good correlations of the absorptions at 328 nm and 344 nm can be found for LP-DOAS observations as well as for both MAX-DOAS data sets. For MAX-DOAS different settings for the spectral retrieval were tested and yielded similar results, however, for the larger fit interval slightly larger (+10%) absorption cross-section peak values are estimated. The low correlation for the setting not using the Taylor expansion approach by Puķīte et al. (2010) on the large fit interval was expected as here strong ozone absorption can produce significant residual structures, which can interfere with other fitted species.

### 4.2.3 Uncertainties

For PPAO, the distribution of fit errors of the $O_4$ absorption at 328 nm has its maximum at $5 \times 10^{41}$ molec$^2$ cm$^{-5}$ (similar as in Figure 4, corresponds to a peak OD of $4.6 \times 10^{-5}$ using the shifted absorption peak from 344 nm), while the distribution of $O_4$ dSCDs at 328 nm has its maximum at $6 \times 10^{42}$ molec cm$^{-2}$. Thus most observations are above the detection limit.



| | | Lowest wavelength [nm] | Settings | $R^2$ | n | $\sigma_{328\,nm}$ [$10^{-47}$ cm$^5$ molec$^{-2}$] |
|---|---|---|---|---|---|---|
| 1 | Antarctica (LP-DOAS) | 320.0 | | 0.56 | 1921 | $1.69 \pm 0.15$ |
| 2 | | 322.5 | | 0.76 | 23573 | $1.59 \pm 0.01$ |
| 3 | | 322.5 | + VRS | 0.73 | 23597 | $1.55 \pm 0.01$ |
| 4 | PPAO (MAX-DOAS) | 322.5 | no Taylor expansion | 0.66 | 22491 | $1.76 \pm 0.01$ |
| 5 | | 311.5 | | 0.90 | 20297 | $1.76 \pm 0.02$ |
| 6 | | 311.5 | no Taylor expansion | 0.29 | 18090 | $1.94 \pm 0.03$ |
| 7 | Polarstern (MAX-DOAS) | 322.5 | | 0.68 | 18888 | $1.47 \pm 0.01$ |

Rows 2–6 bracketed: $1.72 \pm 0.17$

**Table 3.** Results for the O$_4$ absorption at 328nm for different fit ranges and settings. MAX-DOAS data was selected according to RMS $< 4 \times 10^{-4}$, LP-DOAS data below an RMS of $1.2 \times 10^{-4}$, which then results in a different value for n, the number of valid observations. The correlation coefficient $R^2$ was calculated in each case. The peak magnitude of the absorption cross-section at 328 nm was calculated using the O$_4$ cross-section published by Thalman and Volkamer (2013) using the maximum cross-section value of $9.5 \times 10^{-47}$ cm$^5$ molec$^{-2}$ at 344.0 nm.

The influence of strong ozone absorption is largely compensated for by the Taylor expansion approach and even allows fits at sufficiently low RMS down to 311.5 nm. The deduced magnitude of the O$_4$ absorption at 328 nm is slightly larger for larger fit intervals, compare Table 3.

The effect of VRS is negligible especially at the lower end of the fit interval, as can be also seen from Figure 4. It is however correlated with the Ring signal as previously reported for the blue spectral range and increases slightly the number of valid observations while the deduced magnitude of the O$_4$ absorption stays constant within the fit error.

The difference in air mass factors of O$_4$ at low elevation angles at wavelengths of 328 nm and 344 nm is expected to lead to an underestimation of the absolute O$_4$ absorption cross-section at 328 nm in MAX-DOAS observations using the approach presented above. Using a set of 10 representative aerosol profiles with aerosol optical thicknesses ranging from 0–5 and simulating the resulting O$_4$ dSCDs at 328 nm and 344 nm using SCIATRAN (Rozanov et al., 2014) yields an underestimation of 14%. This is slightly less than one could have expected for the pure Rayleigh case, yielding $\approx 1 - (328/343)^4 = 16\%$. Applying this to the observed data from Table 3, this means that either the LP-DOAS values underestimate or the MAX-DOAS results overestimate the real ratio. Estimating the potential systematic measurement errors by the measurement error itself, LP- and MAX-DOAS results would however still agree with each other, as the measurement errors and therefore the scatter of data points are of the same magnitude as the expected difference between the two DOAS measurement types.





### 4.3 Absorption band at 419 nm

For a large $O_4$ dSCD of $1 \times 10^{44}$ molec$^2$ cm$^{-5}$ and using the ratio of the magnitudes of the other $^3\Sigma^3\Sigma \rightarrow^1 \Delta^1\Sigma$ absorption bands at 446 nm and 477 nm of about 12.7 (Thalman and Volkamer, 2013), the peak magnitude of the $O_4$ absorption at 419 nm could be expected to amount to $4 \times 10^{-4}$. It is however unclear, if this extrapolation is valid. For the $^3\Sigma^3\Sigma \rightarrow^1 \Delta^1\Delta$ absorption bands at 361, 344 and 329 nm this is approximately the case as seen in subsection 4.2.

The absorption structure reported by Ellis and Kneser (1933) around 419 nm is overlayed by water vapour absorption at around 416 nm with a peak absorption of $3 \times 10^{-3}$ for a water vapour dSCD of $4 \times 10^{23}$ molec cm$^{-2}$ at a spectral resolution of 0.5 nm using the HITRAN2012 line list (Rothman et al., 2013). From MAX-DOAS and LP-DOAS observations it was reported that these absorption lines are overestimated by a factor of two in HITRAN2012 (Lampel et al., 2015b), while the overall shape is relatively well reproduced. Thus the water vapour absorption contains some uncertainty, which could have an effect on the detection of the $O_4$ absorption band at 419 nm. This is the main reason why the wavelength of the maximum $O_4$ absorption around 419 nm is difficult to estimate.

Therefore the absorption band at 419 nm was fitted in different wavelength intervals, including and excluding the larger water vapour absorption peaks around 442 nm. The variation of the absolute $O_4$ absorption at 419 nm in these two intervals was less than 25%.

To circumvent the potential influence of water vapour absorption in this spectral region, an overdetermined system of linear equations was set up in order to quantify the contribution of water vapour as well as $O_4$ absorption on the apparent $O_4$ absorption at 419 nm. The 'true' dSCDs $S$ were determined at a wavelength close to the wavelength of interest, in this case for the water vapour absorption bands and the $O_4$ absorption band at around 477 nm. These are both more than 10 times stronger than their respective absorptions between 410-420 nm. This can be done as enough variation in water vapour concentrations is found for all MAX-DOAS campaigns included here, thus not leading to linear dependent data points of water vapour and $O_4$ dSCDs.

$$c_{H2O}^{442nm} * S_{H2O} + c_{O4} * S_{O4}^{470nm} = S_{O4}^{419nm} \tag{1}$$

The resulting $c_{O4}$ is $0.0785 \pm 0.007$ or in other words, the absorption peak at 419 nm is $12.7 \pm 1.2$ times smaller than the absorption peak around 445 nm.

The resulting $c_{H2O}$ is $(3.4 \pm 1.0) \times 10^{18}$ molec cm$^{-3}$. This results for a $H_2O$ dSCD of $3 \times 10^{23}$ molec cm$^{-2}$ in a change of the $O_4$ dSCD of $1 \times 10^{42}$ molec$^2$ cm$^{-5}$, i.e. the main variation of the obtained $O_4$ dSCD is indeed caused by the $O_4$ dSCD variation. These show a mean value of $2.9 \times 10^{43}$ molec$^2$ cm$^{-5}$ and a standard deviation of $2.1 \times 10^{43}$ molec$^2$ cm$^{-5}$ at $3^{\circ}$ elevation.

However, as the exact shift of the $O_4$ absorption structure is only poorly restricted from the MAX-DOAS measurements, the remaining uncertainty strongly depends on the position of the absorption peak. Using different fit settings and different positions of the absorption peak ($\pm 0.3$ nm) yields results for $c_{O4}$ within $0.08 \pm 0.05$. 0.12 is actually obtained when ignoring the possible interference with water vapour absorption.



### 4.3.1 LP-DOAS observations

The $O_4$ absorption around 419 nm is close to its detection limit. The maximum of the distribution of $O_4$ fit errors is $7 \times 10^{41}$ molec$^2$ cm$^{-5}$, the maximum of the distribution of $O_4$ dSCDs is $6 \times 10^{42}$ molec$^2$ cm$^{-5}$.

Using the available data sets a reliable conclusion on the magnitude of the $O_4$ absorption is difficult to draw. With an estimated value of $0.07 \pm 0.05$ the ratio of magnitudes to the next larger $O_4$ absorption peak seems to be similarly as the ratio of the magnitudes of the other $^3\Sigma^3\Sigma \rightarrow^1 \Delta^1\Sigma$ absorption bands at 446 nm and 477 nm of about $0.079$ (Thalman and Volkamer, 2013).

## 5 Conclusions

We analysed atmospheric measurements of LP- and MAX-DOAS setups from different field campaigns in order to estimate the magnitude and wavelength of previously reported $O_4$ absorption peaks at 328 nm and 419 nm, for which no spectrally ressolved literature cross-sections are currently available and which have not been reported from atmospheric observations so far.

The main conclusion is that both $O_4$ absorption peaks at 328 nm and 419 nm can be observed using current MAX-DOAS setups and therefore have the potential to introduce biases in the spectral retrieval of weak absorbers. Further laboratory studies are needed quantify the magnitude of these small absorption peaks.

The $O_4$ absorption peak at 328nm was unambiguously identified. Its magnitude agrees with a previous publication by Salow and Steiner (1936) and is found to be $0.18 \pm 0.02$ of the magnitude of the next absorption peak at 344 nm using LP and MAX-DOAS observations. This results in a maximum peak absorption cross-section based on Thalman and Volkamer (2013) of $(1.7 \pm 0.2) \times 10^{-47}$ cm$^5$ molec$^{-2}$ at $328.51 \pm 0.15$ nm. The impact on incoming sun radiation in the spectral region from 323–331 nm is small and amounts to $7.8 \times 10^{-4}$ Wm$^{-1}$ for zenith sun and an $O_4$ VCD of $1.3 \times 10^{43}$ molec$^2$ cm$^{-5}$ using the solar atlas by Chance and Kurucz (2010).

It is interesting to note that the potential $SO_2$ oxidation by singlet oxygen (Sorokin, 2010) requires singlet oxygen molecules with the vibrational eigenstate $\nu > 2$. In particular these weak absorption near 328 nm (and near 315) leads to the formation of singlet oxygen molecules in the 3rd (and 4th) vibrationally excited state (see Table 2) and thus could play a role for $SO_2$ oxidation in the atmosphere.

The impact on trace gas retrievals depends on the fit settings of the respective trace gas and instrumental properties. For DOAS measurements in pristine to semi-polluted regions, a significant impact is expected for spectral retrievals of HCHO, BrO, $SO_2$ and OClO, which could encompass this spectral region. Previous publications often avoided this spectral region. We suggest that the reason for the previously observed discrepancies was often rather the $O_4$ absorption which was not accounted for rather than the increasing influence of tropospheric and stratospheric ozone absorption towards shorter wavelengths.

Incorporating this $O_4$ absorption in the spectral retrievals of the above-mentioned absorbers will lead to a substantial improvement of the respective detection limits as additional absorption bands can be included in the spectral retrieval. In our evaluations, extending the fit range lower limit from 332.5 to 322.5 nm led to a reduction of the fit error by $\approx 35\%$ for HCHO



and BrO. It furthermore significantly reduced the previously observed interferences between the BrO and HCHO absorption structures (Pinardi et al., 2013).

The $O_4$ absorption peak at 419nm was difficult to identify using the method presented here, as it is difficult to exclude the possible influence of water vapour absorption, which overlays the $O_4$ absorption structure and is rather poorly constrained (Lampel et al., 2015b). Its magnitude is estimated to be about $0.07 \pm 0.05$ of the absorption peaks at 446 nm. No published data is available for the absorption at 419 nm. Based on Thalman and Volkamer (2013), it results in a peak absorption cross-section at $419.0 \pm 0.4$ nm of $(3.7 \pm 0.2.7) \times 10^{-48}$ cm$^5$ molec$^{-2}$.

Also in the case of the 419 nm absorption, the impact on trace gas retrievals depends on the fit settings for the respective trace gas and instrument parameters, but an influence can be expected for the spectral retrievals of weak absorbers in pristine regions, such as IO and $NO_2$.

However, the $O_4$ absorption peak at around 419 nm cannot explain the observed differences between different water vapour absorption cross-sections in recent literature (compare Lampel et al. (2015b)), but could contribute to previously observed systematic residual structures.

## Appendix A: Supplement

According to the procedure described above, a merged absorption cross-section of $O_4$ based on Thalman and Volkamer (2013) was calculated. The absorption peak at 344 nm was shifted by $+1366$cm$^{-1}$ to shorter wavelengths and scaled by $0.17$ according to Table 3. These cross-section values were then added to the original $O_4$ absorption cross-section at 293K below 331 nm. To avoid negative cross-section values below 337.5 nm after convolution to instrument resolution, absorption cross-section values between 331 and 337.5 nm were set to zero. The resulting file is provided as a supplement. As the magnitude of the $O_4$ absorption peak around 419 nm remains uncertain, this procedure was not repeated in the blue spectral range.

*Acknowledgements.* We thank Mingxi Yang and coworkers for operating the MAX-DOAS instrument at the Penlee Point Atmospheric Observatory. We also thank Timothy Hay for the help in performing the LP-DOAS measurements and NIWA and Antarctica NZ for hosting our campaign (K084) and the received support. We thank the captain, officers and crew of RV Polarstern for support during cruise ANT XXVIII. Especially for the support by J. Rogenhagen/FIELAX/AWI and technicians on board. We thank Jan-Marcus Nasse for doing maintenance of the MAX-DOAS on Polarstern in the shipping yard before the campaign listed above.





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
