# Peer review of "Detection of $O_4$ absorption around 328 nm and 419 nm in measured atmospheric absorption spectra"

_Atmospheric Chemistry and Physics, 2017_

## Author Comment (AC1) · 11 Sep 2017

R. G. Ryan pointed out that the previous supplement file contained erroneous data, which is why the supplement file has been replaced today by a corrected one. We thank Robert for pointing this out.

---

## Referee Comment (RC1) · R. Thalman (Referee) · 15 Sep 2017

General Comments

Lampel and co-authors present evidence of oxygen O2-O2 collision induced absorption bands at 328 and 419 nm from a combined analysis of Multi Axis (MAX) and Long Path (LP) Differential Optical Absorption Spectroscopy (DOAS) field measurements. The work presents analysis of the effects of these minor absorption bands on the retrieval of other absorbers (HCHO, BrO, H2O) in both MAX and LP-DOAS applications. The analysis utilizes high signal to noise retrievals to identify the band strengths in air to constrain interference with other fitted absorbers. This is especially true with re-

gard to the application to satellite retrievals. The paper is well written and presents a compelling analysis that should be considered in trace gas measurements as well as motivate further laboratory studies to constrain these weak absorption bands. I recommend publication after a few minor corrections as noted below.

Specific Comments:

The report of the cross-section value for the 419 nm band is only found in the Abstract, the estimate should also be included in the results. While the 419 nm band value in the analysis is definitely uncertain (a point well made by the authors), this should still be reported in the body of the paper.

Technical Corrections:

Page 2 line 21: Missing reference (?? in parenthesis).

Page 2 line 23-24: The Vogel 2013 is referenced as the retrieval method for the earlier 2012 paper, is this correct?

Page 3 line 26: reword as "The spectrometer, an Acton 300i,"

Page 4 line 3: This paragraph needs some work (as follows:)

Page 14 line 31: "is obtained when ignoring..." (remove the first actually).

Page 15 line 1: molec2 cm-5 goes into the margin.

Page 15 line 11: resolved (one s).

Page 15 line 29: "was likely the O4 absorption which..." Drop the first rather.

---

## Referee Comment (RC2) · H. Finkenzeller (Referee) · 11 Oct 2017

general comments:

This study applies regression analysis to data sets of atmospheric spectra to find evidence for the incompleteness of currently available oxygen associated absorption cross sections around 328nm and 419nm. Specifically, no spectrally resolved absorption cross sections are available for these bands. While the study does not claim to generate accurate new cross sections, it gives approximate values for the peak absorption strength and peak wavelength for the bands at 328nm and 419nm. The data sets and methods used seem to be well suited for the purpose of the study. Wavelength

ranges are motivated from earlier studies. Band shapes are transferred from neighboring bands in a zeroth order approximation. I do not see a significant potential to extract more information from the data sets for the purpose of the study by using additional or other methods than those presented in the study. The study is a helpful contribution moving the remote sensing technique forward, underlining that better knowledge about oxygen collision induced absorption cross sections is necessary to overcome current limitations. I recommend publication after a few minor adjustments, as described below.

specific comments:

Given the scope of the article, the publications by Dianov-Klokov 1959, 1963, 1964, and 1965 seem to be relevant and should be included in the publication to give the reader a more complete overview of the available literature. The findings in this publication about absorption strengths (of liquid oxygen) are in agreement with the findings presented here.

The assignment of absorption bands to transitions (e.g. in table 2) seems to be incorrect, following e.g. Salow and Steiner 1936, Dianov-Klokov 1959, or the mentioned Greenblatt et al 1990 reference. E.g., for the 328nm band, the initial state is $^3\Sigma_g^- \& ^3\Sigma_g^-$, and the final state is $^1\Sigma_g^+ \& ^1\Sigma_g^+, \nu = 3$. When updating the states throughout the paper, I would recommend to use the full state description for the sake of completeness.

Figure 3: The figure, respectively the analysis, would benefit from an additional fit where the offset is constrained to 0 (assuming the same light path at 328nm and 343nm).

The wording of the article is sometimes not ideal. While not essential for publication, it would benefit from being revised and fine tuned by a native English speaker. The comments below pertain to obvious glitches.

technical corrections/suggestions:

Interactive
comment

page 2, line 16: extra parentheses, remove

page 2, line 21: incorrectly formatted reference

page 3, line 6: rephrase "play a role" to "are responsible for the observed structures" for more clarity

page 3, line 12: "... a measurement campaign in Antarctica was undertaken based at the New Zealand station Scott Base." Rephrase for more clarity.

page 10, line 1: rephrase "As can be seen, ... is visible.", and the following sentence

page 15, line 29: correct "rather ... rather than"
* * *

---

## Author Comment (AC2) · 21 Nov 2017

**Answers to discussion contributions: Detection of $O_4$ absorption around 328 nm and 419 nm in measured atmospheric absorption spectra**

Johannes Lampel[1,*], Johannes Zielcke[2], Stefan Schmitt[2], Denis Pöhler[2], Udo Frieß[2], Ulrich Platt[2], and Thomas Wagner[1]

[1]Max Planck Institute for Chemistry, Mainz, Germany
[2]Institute of Environmental Physics, University of Heidelberg, Germany
[*]Now at: Institute of Environmental Physics, University of Heidelberg, Germany

November 15, 2017

**1 Introduction and general comments**

**We would like to thank the editor Rainer Volkamer and the two reviewers Ryan Thalman and Henning Finkenzeller for their comments, suggestions and corrections. Robert R. Ryan commented offline to the manuscript. All comments together improved greatly the discussion manuscript. We considered each of the points, which are answered below together with a description of the related changes to the manuscript. When reprocessing the data for the supplement of the manuscript, I realized that the relative shifts mentioned in Section 3.3 were slightly wrong due to an inconsistency in using air/vacuum wavelength calibrations for the shifted hypothetical $O_4$ absorption cross-sections. This is fixed now, the respective peak absorption cross-section values in the manuscript and the supplement file are updated. Furthermore I included now the estimate for the absorption peak of $O_4$ at around 419 nm in the supplement file as well, despite the large error of the estimate. Here the radiative transfer effect on the absolute magnitude due to the wavelength dependency of the AMF was previously not considered, but it is now.**

(Numbers of equations, figures, lines and pages refer to the discussion manuscript, if not mentioned otherwise. **Authors' reponses are written in bold face**, the referees' text is shown in normal face.)

**2 Review #1**

Lampel and co-authors present evidence of oxygen O2-O2 collision induced absorption bands at 328 and 419 nm from a combined analysis of Multi Axis (MAX) and Long Path (LP) Differential Optical Absorption Spectroscopy (DOAS) field measurements. The work presents analysis of the effects of these minor absorption bands on the retrieval of other absorbers (HCHO, BrO, H2O) in both MAX and LP-DOAS applications. The analysis utilizes high signal to noise retrievals to identify the band strengths in air to constrain interference with other fitted absorbers. This is especially true with regard to the application to satellite retrievals. The paper is well written and presents a compelling analysis that should be considered in trace gas measurements as well as motivate further laboratory studies to constrain these weak absorption bands. I recommend publication after a few minor corrections as noted below.

**We thank the reviewer for the overall positive assessement.**

**2.1 Specific comments**

1. The report of the cross-section value for the 419 nm band is only found in the Abstract, the estimate should also be included in the results. While the 419 nm band value in the analysis is definitely uncertain (a point well made by the authors), this should still be reported in the body of the paper.

    **We added the following sentence to the end of the paragraph about the observations around 419 nm: 'This then results in a peak value of the $O_4$ absorption cross-section of $(3.7 \pm 2.7) \times 10^{-48}$ $cm^5$ $molec^{-2}$ at a wavelength of $419.02 \pm 0.42$.'.**

**2.2 Technical corrections**

1. Page 2 line 21: Missing reference (?? in parenthesis).

    **?? is now replaced by [Wang et al., 2017].**

2. Page 2 line 23-24: The Vogel 2013 ([Vogel et al., 2013]) is referenced as the retrieval method for the earlier 2012 ([Vogel, 2012]) paper, is this correct?

    **yes, this was the intended meaning of the sentence. The PhD-thesis ([Vogel, 2012]) was finished earlier than the final version of [Vogel et al., 2013]. The observations which led to the restriction of the fit range in [Vogel, 2012] were not included in [Vogel et al., 2013]. We restructured the sentence to make this clearer. It is now:**

    **In addition to that, for the spectral retrieval of BrO, [Vogel, 2012] found a lower limit for the choice of reliable fit intervals at about 330.6 nm, which could have been caused by the neglected $O_4$ absorption at around 328 nm - the FWHM (Full Width Half Maximum) of the known absorption peaks at 344 and 360 nm amounts to 4 nm. The method from [Vogel et al., 2013] was applied here on measured atmospheric spectra along different light path lengths.**

3. Page 3 line 26: reword as The spectrometer, an Acton 300i,

   **done**

4. Page 4 line 3: This paragraph needs some work (as follows:)

   **Thanks for pointing this out. It now reads as follows:**

   **The measurement procedure was as follows: Spectra were acquired alternatingly on both light paths. For each light path, spectra were recorded at four different wavelength intervals. The region of interest for this study is between 271 nm and 355 nm. For each of those regions, 25 spectra with a saturation of 70% were recorded, with a maximum duration of 1 s per spectrum. Typical exposure times for one spectrum were between 6 ms and 500 ms depending on the wavelength range, visibility conditions and the used arc lamp.**

5. Page 14 line 31: is obtained when ignoring. . . (remove the first actually).

   **done**

6. Page 15 line 1: molec2 cm-5 goes into the margin.

   **done**

7. Page 15 line 11: resolved (one s).

   **fixed**

8. Page 15 line 29: was likely the O4 absorption which. . . Drop the first rather.

   **fixed**

**3  Review #2**

This study applies regression analysis to data sets of atmospheric spectra to find evidence for the incompleteness of currently available oxygen associated absorption cross sections around 328nm and 419nm. Specifically, no spectrally resolved absorption cross sections are available for these bands. While the study does not claim to generate accurate new cross sections, it gives approximate values for the peak absorption strength and peak wavelength for the bands at 328nm and 419nm. The data sets and methods used seem to be well suited for the purpose of the study. Wavelength ranges are motivated from earlier studies. Band shapes are transferred from neighboring bands in a zeroth order approximation. I do not see a significant potential to extract more information from the data sets for the purpose of the study by using additional or other methods than those presented in the study. The study is a helpful contribution moving the remote sensing technique forward, underlining that better knowledge about oxygen collision induced absorption cross sections is necessary to overcome current limitations. I recommend publication after a few minor adjustments, as described below.

**We thank Henning Finkenzeller for his evaluation and implemented most of his suggestions in the revised manuscript. We like to point out that the extraction of more detailed spectral information of the**

absorption bands could be possible by regression analysis of residual spectra on a channel-by-channel basis. However, due to the low signal-to-noise ratio and the number of overlaying absorbers these attempts failed during the preparation of the manuscript.

**3.1 Specific comments:**

1. Given the scope of the article, the publications by Dianov-Klokov 1959, 1963, 1964, and 1965 seem to be relevant and should be included in the publication to give the reader a more complete overview of the available literature. The findings in this publication about absorption strengths (of liquid oxygen) are in agreement with the findings presented here.

   **Thanks for pointing this out, we now also cite [Dianov-Klokov, 1959, Dianov-Klokov, 1964] in the revised manuscript.**

2. The assignment of absorption bands to transitions (e.g. in table 2) seems to be incorrect, following e.g. Salow and Steiner 1936, Dianov-Klokov 1959, or the mentioned Greenblatt et al 1990 reference. E.g., for the 328nm band, the initial state is $^3\Sigma_g^-\&^3\Sigma_g^-$ and the final state is $^1\Sigma_g^+\&^1\Sigma_g^+(\nu=3)$ . When updating the states throughout the paper, I would recommend to use the full state description for the sake of completeness.

   **yes, indeed, we made an error and interchanged the $\Sigma$ and $\Delta$s at some point. We fixed this and also used now the full state description in the revised manuscript.**

3. Figure 3: The figure, respectively the analysis, would benefit from an additional fit where the offset is constrained to 0 (assuming the same light path at 328nm and 343nm).

   **We added an additional fit with an offset constrained to zero to the figure and additionally incorporated this information in the text and in table 3. The result of the fit which is constrained to zero for the reference measurements actually fits the MAX-DOAS observations better than the other, previous fit, which was not constrained for the reference measurements. This makes sense as reference measurements have by definition all column densities at zero.**

4. The wording of the article is sometimes not ideal. While not essential for publication, it would benefit from being revised and fine tuned by a native English speaker. The comments below pertain to obvious glitches.

   **We read through the article again and removed some of these. We hope that this is now acceptable. Additionally the article will be copy-edited during the final preparation of the manuscript.**

**3.2 Technical corrections/suggestions:**

1. page 2, line 16: extra parentheses, remove

   **done**

2. page 2, line 21: incorrectly formatted reference

   **fixed**

3. page 3, line 6: rephrase "play a role" to "are responsible for the observed structures" for more clarity

   **We replaced it by 'are responsible for the observed absorption structures'.**

4. page 3, line 12: "... a measurement campaign in Antarctica was undertaken based at the New Zealand station Scott Base." Rephrase for more clarity.

   **We now write 'In 2012, a measurement campaign was undertaken based at New Zealand's research station Scott Base in Antarctica.'**

5. page 10, line 1: rephrase "As can be seen, ... is visible.", and the following sentence

   **We removed 'As can be seen' in the revised manuscript.**

6. page 15, line 29: correct "rather ... rather than"

   **Nothing to correct as far as I can see.**

**References**

[Dianov-Klokov, 1959] Dianov-Klokov, V. I. (1959). On the question of the origion of the spectrum of liquid and compressed oxygen (12.600 - 3.000 å). *Opt. Spectrosc.*, 6:290–293.

[Dianov-Klokov, 1964] Dianov-Klokov, V. I. (1964). Absorption spectrum of condensed oxygen in the 1.26-0.3 $\mu$m region. *Opt. Spectrosc.*, 20:530–534.

[Vogel, 2012] Vogel, L. (2012). *Volcanic plumes: Evaluation of spectroscopic measurements, early detection, and bromine chemistry.* Dissertation, Institut für Umweltphysik, Heidelberg University.

[Vogel et al., 2013] Vogel, L., Sihler, H., Lampel, J., Wagner, T., and Platt, U. (2013). Retrieval interval mapping: a tool to visualize the impact of the spectral retrieval range on differential optical absorption spectroscopy evaluations. *Atmospheric Measurement Techniques*, 6(2):275–299.

[Wang et al., 2017] Wang, Y., Beirle, S., Hendrick, F., Hilboll, A., Jin, J., Kyuberis, A. A., Lampel, J., Li, A., Luo, Y., Lodi, L., Ma, J., Navarro, M., Ortega, I., Peters, E., Polyansky, O. L., Remmers, J., Richter, A., Rodriguez, O. P., Roozendael, M. V., Seyler, A., Tennyson, J., Volkamer, R., Xie, P., Zobov, N. F., and Wagner, T. (2017). Max-doas measurements of hono slant column densities during the madcat campaign: inter-comparison and sensitivity studies on spectral analysis settings. *Atmospheric Measurement Techniques Discussions*, 2017:1–38.

---

## Author Response (AR2)

**Co-editor comment: Minor revisions for Lampel et al AMTD 2017 'Detection of O4 absorption around 328 nm and 419 nm in measured atmospheric absorption spectra'**

Comments to the Author:
The authors have responded to all reviewer comments, and the revised manuscript is almost ready for publication.

**We thank the co-editor Rainer Volkamer for his suggestions for the manuscript on O4 absorption around 328 and 419nm. We addressed each of the points in the revised manuscript. Our answers and comments are written in bold face here.**

The addition of references by Dianov-Klokov (1959, 1964) (reviewer#2) provides an independent data set to compare the central wavelengths, and relative intensities determined in this work. The revised text now cites these papers, and refers to Table 2 for an overview of available literature values. However, Table 2 of the revised manuscript does not include the actual values from these important references. How do they compare? A few sentences discussing whether the rel int in liquid oxygen actually compare with those in the gas-phase would be valuable, but such discussion is currently missing. Can these data be used to reduce the uncertainty of the estimated cross-sections?

**The values obtained for the 328nm absorption band were compared in Figure 5 to those of Salow and Steiner, who measured these absorptions in the gaseous phase. We added a sentence each comparing the results for both absorption bands to those from Salow and Steiner and also to Dianov-Klokov. For the 419nm peak we added a concluding sentence comparing the results to those from Dianov-Klokov.**

**A clear difference of the relative magnitude for the absorption of O4 in the liquid and in the gas phase cannot be seen from the available data from Salow and Steiner, Ellis and Kneser and Dianov-Klokov. We added this to the revised manuscript. Their relatively wide-spread values cannot be used to restrain the results for the estimated cross-section.**

Finally, any recommendations for future laboratory work that might arise from this comparison should be spelled out.

**We already wrote that these results show that there is the need of additional laboratory studies. We suggest now additionally that these could be combined with measurements of weak water vapour absorption structures in the blue around 419nm as well as around 363nm, as the weak O4 absorption features are of similar magnitude as the relatively uncertrain weak water vapour absorption structures.**

On a more technical note, the individual columns in the section 'rel int' of Table 2 currently do not yet use common units. Only the relative intensity matters in context of the presented work. A common set of units will make the comparison of earlier studies more direct, and help inform the above discussion.

**I normalized the relative intensities in the table to the absorption peak at 446 and 344nm, respectively to simplify direct comparisons. As for the Dianov-Klokov publications, I did not find the magnitude of the absorption peaks as numeric values in a table, therefore I extracted them from one of the figures. This is also mentioned now in the caption of the table, which now contains the wavelengths and relative intensity values from the Dianov-Klokov publications for liquid oxygen.**

Please check and use a consistent number of significant figures for reporting the cross-section values and their uncertainties (e.g., see abstract, and throughout the paper).

**I checked this and found that e.g. the values in the abstract and in the overview table 3 differ, as the table lists the direct results from the correlation while the abstract shows the number which results after correcting for the shorter light path at 328nm compared to 344nm which would have resulted in an underestimation of the magnitude of the O4 absorption peak. I added here a sentence in the paragraph summarizing the results of the table. The caption of the table already contained a note that the values in the table are not corrected for the wavelength dependence of the O4 AMF.**

These changes should be straightforward, and will benefit the final quality of the paper.

**Thanks for pointing these out. We hope that now all of these suggestions have been addressed satisfyingly.**

[revised manuscript text omitted]